# Rethinking Node-wise Propagation for Large-scale Graph Learning

## ABSTRACT

Scalable graph neural networks (GNNs) have emerged as a promising technique, which exhibits superior predictive performance and high running efficiency across numerous large-scale graph-based web applications. However, (i) Most scalable GNNs tend to treat all nodes in graphs with the same propagation rules, neglecting their topological uniqueness; (ii) Existing node-wise propagation optimization strategies are insufficient on web-scale graphs with intricate topology, where a full portrayal of nodes' local properties is required. Intuitively, different nodes in web-scale graphs possess distinct topological roles, and therefore propagating them indiscriminately or neglect local contexts may compromise the quality of node representations. This intricate topology in web-scale graphs cannot be matched by small-scale scenarios. To address the above issues, we propose **A**daptive **T**opology-aware **P**ropagation (ATP), which reduces potential high-bias propagation and extracts structural patterns of each node in a scalable manner to improve running efficiency and predictive performance. Remarkably, ATP is crafted to be a plug-and-play node-wise propagation optimization strategy, allowing for offline execution independent of the graph learning process in a new perspective. Therefore, this approach can be seamlessly integrated into most scalable GNNs while remain orthogonal to existing node-wise propagation optimization strategies. Extensive experiments on 12 datasets, including the most representative large-scale ogbn-papers100M, have demonstrated the effectiveness of ATP. Specifically, ATP has proven to be efficient in improving the performance of prevalent scalable GNNs for semi-supervised node classification while addressing redundant computational costs.

## CCS CONCEPTS

• **Computing methodologies** → **Semi-supervised learning settings**; **Neural networks**.

## KEYWORDS

Graph Neural Networks; Scalability; Semi-Supervised Learning

**ACM Reference Format:**
Anonymous Author(s). 2018. Rethinking Node-wise Propagation for Large-scale Graph Learning. In *Proceedings of Make sure to enter the correct conference title from your rights confirmation emai (Conference acronym 'XX).* ACM, New York, NY, USA, 14 pages. https://doi.org/XXXXXXX.XXXXXXX

## 1 INTRODUCTION

Recently, the rapid growth of web-scale graph mining applications has driven needs for efficient analysis tools to tackle scalability challenges in the real world, including social analysis [45, 72, 73] and e-commerce recommendations [6, 59, 61]. Scalable graph neural networks (GNNs), as a new machine learning paradigm for large-scale graphs, have inspire significant interests due to their higher efficiency than vanilla GNNs in node-level [31, 57, 71], edge-level [5, 50, 68], and graph-level [52, 62, 74] downstream tasks.

Fundamentally, the core of GNN's scalability lies in the simplified aggregators or weight-free deep structural encoding. Therefore, existing scalable GNNs fall into two categories: (i) Sampling-based methods [13, 17, 28, 33, 67] employ well-designed strategies to select suitable graph elements (e.g. nodes or edges) for computation-friendly message aggregators. Although they are effective, these approaches are imperfect because they still face high communication costs in sampling and the sampling quality highly influences the performance. As a result, many recent advancements achieve scalability by decoupling paradigm, orthogonal to the sampling technologies. (ii) Decouple-based methods [14, 24, 26, 53, 76] treat weight-free feature propagation as pre-process and combine propagated results with reasonable learnable architectures to achieve efficient training. For example, SGC [57] combines propagated node features with simple linear regression and achieves performance comparable to carefully designed GNNs. In weight-free feature propagation, neighbors' features are iteratively combined with the current node. This relies on the homophily assumption [43, 48, 58], where connected nodes share similar features and labels, thereby helping predict node labels. We refer to the feature propagation that does not distinguish between nodes as graph propagation.

Despite their effectiveness, most of the aforementioned scalable GNNs fail to consider the unique roles played by each node in the topology. Instead, they employ fixed propagation rules for computation. Therefore, there is still room for refining the granularity of graph propagation. To improve it, NDLS [70] proposes node-wise propagation (NP), which quantifies the difference between the current propagated node features and the theoretically over-smoothed node features obtained by infinite step propagation. This enables custom propagation steps for each node. Building upon this concept of NP, NDM [32] introduces an extra power parameter to extend the graph heat diffusion function DGC [55], separating the terminal time from the propagation steps for each node. SCARA [41] further extends NP by the node feature-push operations, achieving attribute mining for each node. Despite offering practical NP strategies, these methods rely on spectral analysis and the generalized steady-state distribution of the fixed propagation operator to customize the rigid NP strategy from a global perspective. Therefore, these methods often yield high-bias results due to over-reliance on the coarse propagation operator in web-scale graphs with intricate topology. Meanwhile, real-world node classification on web-scale

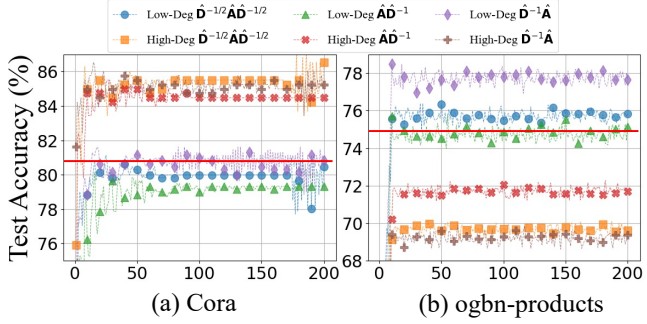

**Figure 1: Performance in the Cora (2.7k nodes) and ogbn-products (2449k nodes). The x-axis is the training epoch. The red line denotes the baseline performance for all nodes.**

graphs with intricate topology heavily relies on the local node context (LNC), which refers to a general characterization of nodes based on their *features*, *positions* in the graph, and *local topological structure*. Regrettably, existing methods ignore this crucial factor.

To further illustrate, we utilize the node degree to represent the LNC, which directly influences the local connectivity of nodes. Specifically, in Cora and ogbn-products, we classify nodes with degrees less than or equal to 3 and 5 as Low-Deg and other nodes as High-Deg, where Low-Deg at the graph's periphery with fewer connections and High-Deg located at the center of densely connected communities. Subsequently, in Fig. 1, we use various propagation operators combined with 3-layer SGC to evaluate the predictive performance of nodes with different LNC (i.e. node degrees) in these two datasets. The related notations can be referred to Sec. 2.1.

Intuitively, different propagation operators capture knowledge based on nodes' LNC from distinct perspectives during message passing , resulting in the different node representations for classification: (i) the symmetric normalization propagation operator $\hat{\mathbf{D}}^{-1/2}\hat{\mathbf{A}}\hat{\mathbf{D}}^{-1/2}$ [38] considers both current node and neighbors' LNC to perform unbiased message passing; (ii) the random walk-based propagation operator $\hat{\mathbf{D}}^{-1}\hat{\mathbf{A}}$ [67] only considers current node LNC, leading to a more inclusive knowledge acquisition from its neighbors without additional normalization; (iii) the reverse random walk-based operator $\hat{\mathbf{A}}\hat{\mathbf{D}}^{-1}$ [60] only considers neighbors' LNC, enhancing the capacity to discriminate between neighbors to achieve fine-grained message aggregation. The following analysis illustrates two key insights acquired through examining experimental results.

**Key Insight 1**: *From the global perspective, we need to focus on High-Deg in web-scale scenarios to mitigate the negative impacts of high-bias propagation to ensure consistent performance.* As depicted in Fig. 1, we observe that the Low-Deg performance remains consistent (same trends in three operators) and stable (similar performance to the red baseline) across two datasets. In contrast, the inconsistent and unstable High-Deg performance has prompted us to conduct a more in-depth analysis with different graph scales.

Research on complex networks [16, 20] indicates that the topology of large-scale graphs is highly intricate, which results in the emergence of densely connected communities with indiscernible High-Deg. Consequently, in ogbn-products, densely connected communities possess more intricate and ambiguous LNC, misleading High-Deg during graph propagation (i.e. high-bias propagation). This explains why considering the LNC of neighbors through $\hat{\mathbf{A}}\hat{\mathbf{D}}^{-1}$

can yield better High-Deg performance but worse than baseline. In contrast, the topology of small-scale Cora is relatively straightforward, enabling High-Deg to outperform the baseline by aggregating more favorable messages. This explains why $\hat{\mathbf{D}}^{-1/2}\hat{\mathbf{A}}\hat{\mathbf{D}}^{-1/2}$ is beneficial for predicting High-Deg, where both the knowledge of current node and its neighbors hold equal significance.

**Key Insight 2**: *From the local perspective, leveraging appropriate propagation operators across different scenarios to effectively capture relevant LNC can improve node predictive performance.* After analyzing High-Deg in graphs of different scales, we conduct a thorough examination of the roles played by different propagation operators in consistent performance trends observed for Low-Deg. As depicted in small-scale Cora, the success of $\hat{\mathbf{D}}^{-1}\hat{\mathbf{A}}$ in Low-Deg stems from its enhanced focus on aggregating neighbor features, which breaks potential feature sparsity issues caused by fewer neighbors. In contrast, $\hat{\mathbf{D}}^{-1/2}\hat{\mathbf{A}}\hat{\mathbf{D}}^{-1/2}$ and $\hat{\mathbf{A}}\hat{\mathbf{D}}^{-1}$ apply progressively enhanced normalization to propagated messages based on neighbors' LNC, thereby constraining the aggregation of knowledge from Low-Deg neighbors. This is also applicable to large-scale scenarios in Fig. 1(b).

Motivated by the above key insights, in this paper, we propose **A**daptive **T**opology-aware **P**ropagation (ATP), which offers a plug-and-play solution for existing GNNs. Specifically, ATP first identifies potential high-bias propagation through graph propagation analysis and then employs a masking mechanism to regularize the node-wise propagation mechanisms (motivated by Key Insight 1). After that, ATP employs a general encoding approach to represent node-dependent LNC without learning, which is then used to tailor propagation rules for each node (motivated by Key Insight 2).

**Our contributions.** (1) *New Perspective.* To the best of our knowledge, this work is the first to address the adverse impact of intricate topology in web-scale graph mining applications on the semi-supervised node classification paradigm, providing valuable empirical analysis. (2) *New Method.* We propose ATP, a user-friendly and flexible NP optimization strategy. It effectively mitigates the high-bias propagation caused by intricate topology and employs a weight-free approach to represent the LNC of different nodes, thus improving most scalable GNNs. Importantly, ATP is orthogonal to the existing NP optimization strategies. (3) *SOTA Performance.* We conduct experiments on prevalent scalable GNNs and 12 benchmark datasets including the representative large-scale ogbn-papers100M. Empirical results demonstrate that ATP has a significant positive impact on existing scalable GNNs (up to 4.96% higher). Furthermore, when combined with existing NP optimization strategies, it exhibits a complementary effect, resulting in additional performance gains.

## 2 PRELIMINARIES

### 2.1 Problem Formulation

Consider a graph $\mathcal{G} = (\mathcal{V}, \mathcal{E})$ with $|\mathcal{V}| = n$ nodes and $|\mathcal{E}| = m$ edges, the adjacency matrix (including self-loops) is $\hat{\mathbf{A}} \in \mathbb{R}^{n \times n}$, the feature matrix is $\mathbf{X} = \{x_1, \ldots, x_n\}$ in which $x_v \in \mathbb{R}^f$ represents the feature vector of node $v$, and $f$ represents the dimension of the node attributes. Besides, $\mathbf{Y} = \{y_1, \ldots, y_n\}$ is the label matrix, where $y_v \in \mathbb{R}^{|\mathcal{Y}|}$ is a one-hot vector and $|\mathcal{Y}|$ represents the number of the classes. The semi-supervised node classification task is based on the topology of labeled set $\mathcal{V}_L$ and unlabeled set $\mathcal{V}_U$, and the nodes in $\mathcal{V}_U$ are predicted with the model supervised by $\mathcal{V}_L$.

## 2.2 Scalable Graph Neural Networks

Motivated by the spectral graph theory and deep neural networks, GCN [38] simplifies the topology-based convolution operator [4] by the first-order approximation of Chebyshev polynomials [37]. The forward propagation of the $l$-th layer in GCN is formulated as

$$\mathbf{X}^{(l)} = \sigma(\tilde{\mathbf{A}}\mathbf{X}^{(l-1)}\mathbf{W}^{(l)}), \ \tilde{\mathbf{A}} = \hat{\mathbf{D}}^{-1/2}\hat{\mathbf{A}}\hat{\mathbf{D}}^{-1/2}, \tag{1}$$

where $\hat{\mathbf{D}}$ represents the degree matrix of $\hat{\mathbf{A}}$, $\mathbf{W}$ represents the trainable weights, and $\sigma(\cdot)$ represents the non-linear activation function. Intuitively, GCN aggregates the neighbors' representation embeddings from the $(l-1)$-th layer to form the representation of the $l$-th layer. Such a simple paradigm is proved to be effective in various graph-based downstream tasks [25, 35, 64]. However, GCN suffers from severe scalability issues since it executes the feature propagation and transformation recursively and is trained in a full-batch manner. To avoid the recursive neighborhood over expansion, sampling and decouple-based approaches have been investigated.

**Sampling-based methods.** Regarding node-level sampling techniques, GraphSAGE [28] employs random selection to extract a fixed-size set of neighbors for computation within each mini-batch. VR-GCN [13] delves into node variance reduction, achieving size reduction in samples at the expense of additional memory usage. For layer-level sampling, FastGCN [12] proposes importance-based neighbor selection to minimize sampling variance. AS-GCN [33] accounts for correlations among neighbors sampled from upper layers, introducing an adaptive layer-level sampling method for explicit variance reduction. Meanwhile, LADIES [77] adheres to layer constraints, crafting a neighbor-dependent and importance-based sampling approach on a per-layer basis. As for the graph-level sampling strategies, Cluster-GCN [17] initially clusters nodes before extracting nodes from clusters, while GraphSAINT [67] directly samples subgraphs for mini-batch training. GraphCoarsening [34] creates a coarse-grained graph with a reduced number of nodes and edges, derived from the original data. Subsequently, a GNN is trained on this coarser graph and transferred to the original graph for inference. ShaDow [66] first extracts a local subgraph for target entities and employs GNN of arbitrary depth on the subgraph.

**Decouple-based methods.** Recent studies [69] have observed that non-linear feature transformation contributes little to performance as compared to graph propagation. Thus, a new direction for scalable GNN is based on the SGC [57], which reduces GNNs into a linear model operating on $k$-layer propagated features

$$\mathbf{X}^{(k)} = \tilde{\mathbf{A}}^k\mathbf{X}^{(0)}, \ \mathbf{Y} = \text{softmax}\left(\mathbf{W}\mathbf{X}^{(k)}\right), \tag{2}$$

where $\mathbf{X}^{(0)} = \mathbf{X}$ and $\mathbf{X}^{(k)}$ is the $k$-layer propagated features. As the propagated features $\mathbf{X}^{(k)}$ can be precomputed, SGC is easy to scale to large graphs. Inspired by it, SIGN [26] proposes to concatenate the learnable propagated features $\left[\mathbf{X}^{(0)}\mathbf{W}_0, \ldots, \mathbf{X}^{(k)}\mathbf{W}_k\right]$. $S^2$GC [76] proposes to average the propagated results from the perspective of spectral analysis $\mathbf{X}^{(k)} = \sum_{l=0}^k \tilde{\mathbf{A}}^l\mathbf{X}^{(0)}$. GBP [14] utilizes the $\beta$ weighted manner $\mathbf{X}^{(k)} = \sum_{l=0}^k w_l\tilde{\mathbf{A}}^l\mathbf{X}^{(0)}$, $w_l = \beta(1-\beta)^l$. GAMLP [71] achieves information aggregation based on the attention mechanisms $\mathbf{X}^{(k)} = \tilde{\mathbf{A}}^k\mathbf{X}^{(0)} \| \sum_{l=0}^{k-1} w_l\mathbf{X}^{(l)}$, where attention weight $w_l$ has multiple calculation versions. GRAND+ [24] proposes a generalized forward push propagation algorithm to obtain $\tilde{\mathbf{P}}$, which is used to approximate $k$-order PageRank weighted $\tilde{\mathbf{A}}$ with higher flexibility and efficiency. Then it obtains propagated results $\tilde{\mathbf{X}} = \tilde{\mathbf{P}}\mathbf{W}\mathbf{X}^{(0)}$ with $\tilde{\mathbf{P}}$-based data augmentation and learnable $\mathbf{W}$.

**Node-wise Propagation Optimization Strategies.** Despite the aforementioned scalable GNNs utilizing computation-friendly message aggregators (i.e. sampling-based methods) or decoupling paradigms to extend learnable architectures to web-scale graphs with millions of nodes, the majority of existing methods still adhere to fixed propagation rules. This approach, which does not discriminate between nodes, inadvertently overlooks the uniqueness of each node within the topology-based propagation process. Hence, recent studies have introduced fine-grained $k$-step NP optimization strategies in a scalable manner to improve the predictive performance of scalable GNNs for web mining applications. The optimization paradigm can be formally expressed as

$$\tilde{\mathbf{X}} = \sum_{l=0}^k \mathbf{\Pi} \cdot \mathbf{L} \cdot \mathbf{H} \cdot \mathbf{X} [\text{row, col}], \ \mathbf{\Pi} = \sum_{i=0}^l w_i \cdot \left(\hat{\mathbf{D}}^{r-1}\hat{\mathbf{A}}\hat{\mathbf{D}}^{-r}\right)^i,$$

$$\mathbf{L} = \text{Diag}\left\{\mathbb{I}\left[l_u\right] : \forall u \in \mathcal{V}\right\}, \ k = \max\left\{l_u, \forall u \in \mathcal{V}\right\}, \tag{3}$$

$$\mathbf{H} = \frac{\omega^l}{(l!)^\rho \cdot C}, \ C = \sum_{l=0}^\infty \frac{\omega^l}{(l!)^\rho} \leftarrow e^{-\omega}\frac{\omega^l}{l!} \approx e^{-t}\sum_{i=0}^\infty \frac{t^i}{i!},$$

where propagated feature $\tilde{\mathbf{X}}$ is obtained by the various NP optimization perspectives (i.e. $\mathbf{L}$, $\mathbf{H}$, $\mathbf{\Pi}$, and $\mathbf{X}$ [row, col]). In other words, the NP optimization perspectives are diverse. NDLS [70] and NDM [32] customize node-wise propagation step, where $l_u$ represents the propagation step for node $u$, while $\mathbf{L}$ denotes the diagonal matrix composed of indicator vectors $\mathbb{I}$, used to compute the appropriate propagation results, i.e., $\mathbb{I}\left[l_u\right] = 1$ if $l \leq l_u \leq k$ and $\mathbb{I}\left[l_u\right] = 0$ otherwise. As we know, by solving the differential function of Graph Heat Equation $\mathbf{H}$ at time $t$ defined by [18], GDC [27] and DGC [55] obtain the underlying Heat Kernel PageRank parameterized by $\omega$ for fine-grained NP optimization. Notably, we focus solely on describing the heat kernel function used for propagation, omitting the node features $\mathbf{X}$. Additionally, for the sake of reader-friendliness, we present $\mathbf{H}$ and $\mathbf{\Pi}$ in a decoupled manner. Building upon this, NDM introduces normalization factor $C$ and power parameter $\rho$ to improve its expressiveness and generalizability, which can control change tendency for general purposes. Furthermore, SCARA [41] achieves the discovery of potential correlations between nodes by performing fine-grained feature-push operations, transforming the computation entities from $\mathbf{X}$ [row, :] to $\mathbf{X}$ [:, col].

Despite their effectiveness, essential propagation rules are ignored. Specifically, $\mathbf{\Pi}$ is the unified graph propagation equation, which serves as an effective paradigm to model various node proximity measures and basic GNN propagation formulas (i.e. $\tilde{\mathbf{A}}\mathbf{W}^{(l)}$ in GCN and $\tilde{\mathbf{A}}^l$ in SGC). For a given node $u$, a node proximity query yields $\mathbf{\Pi}(v)$ that represents the importance of $v$ with respect to $u$. It captures intricate structural insights from the $l$-hop neighbors, which is guided by weight sequence $w_i$ and probabilities obtained from a $l$-step propagation that originates from a source node $u$ and extends to every node within the graph. More deeply, the propagation kernel coefficient $r \in [0, 1]$ not only affects transport probabilities during the propagation for modeling node proximity but also captures pivotal LNC knowledge detailed in Sec. 1 (i.e. three propagation operators obtained by setting $r = 0$, $r = 1/2$, $r = 1$).

# 3 ATP FRAMEWORK

As a plug-and-play node-wise propagation optimization strategy, the computation of ATP is independent of the graph learning and remains orthogonal to existing NP methods. It commences by employing a masking mechanism for correcting potential high-bias propagation from a global perspective. Then, ATP represents the LNC to custom propagation rules for each node in a weight-free manner from a local perspective. Based on this, ATP serves to curtail redundant computations and provides performance gains by high-bias propagation correction and LNC encoding for existing scalable GNNs. The complete algorithm can be referred to as Algorithm 1.

## 3.1 High-bias Propagation Correction

**Propagation Operator.** For existing GNNs, numerous variations of the Laplacian matrix have been widely employed as propagation operators, where $\mathbf{P} = \hat{\mathbf{D}}^{-1}\hat{\mathbf{A}}$ stands out due to its intuitive and explainable nature. Let $1 = \lambda_1 \geq \lambda_2 \geq \ldots \geq \lambda_n > -1$ be the eigenvalues of $\mathbf{P}$. Suppose the graph is connected, for any initial distribution $\pi_0$, let $\tilde{\pi}(\pi_0) = \lim_{k \to \infty} \pi_0 \mathbf{P}^k$, where $\tilde{\pi}(\pi_0)$ represents the stable state under infinite propagation. Then according to [21], we have $\tilde{\pi}_i = \tilde{\pi}(\pi_0)_i = \frac{1}{n}\sum_{j=1}^{n} \mathbf{P}_{ji}$, where $\tilde{\pi}_i$ is the $i$-th component. If $\mathbf{P}$ is not connected, we can divide $\mathbf{P}$ into connected blocks. Then for each blocks $B_g$, there always be $\tilde{\pi}(\pi_0)_i = \frac{1}{n_g}\sum_{j \in B_g} \mathbf{P}_{ji} \cdot \sum_{j \in B_g}(\pi_0)_j$, where $n_g$ is the number of nodes in $B_g$. To make the following derivation more reader-friendly, we assume $\mathbf{P}$ is connected. Therefore, $\tilde{\pi}$ is independent to $\pi_0$, thus we replace $\tilde{\pi}(\pi_0)$ by $\tilde{\pi}$. To investigate the fine-grained graph propagation, we have the following lemmas

**LEMMA 1.** *The difference between the stable state and $k$-step propagated results represents the upper bound of the convergence rate.*

$$\left|\left(\mathbf{P}^k e_i\right)_j - \tilde{\pi}_j\right| \leq \sqrt{\frac{\tilde{d}_j}{\tilde{d}_i}} \lambda_2^k, \tag{4}$$

*where $\tilde{d}$ denotes the degree of node plus 1 (to include itself by self-loop).*

**LEMMA 2.** *For a graph $\mathcal{G} = (\mathcal{V}, \mathcal{E})$ with the average degree $d_{\mathcal{G}}$, we have $1 - \Delta_{\lambda} = O\left(1/\sqrt{d_{\mathcal{G}}}\right)$, where $\Delta_{\lambda}$ is the spectral gap of $\mathcal{G}$.*

**Global Graph Propagation.** Fundamentally, the core of graph propagation is the trade-off between the node-wise optimal convergence diameters and over-smoothing. This optimal convergence diameter indicates the receptive field required for generating the most effective node representations, whereas exceeding this range would lead to negative impacts due to over-smoothing. While some methods propose node-adaptive $k$ for aggregating valuable information within $k$-hop neighbors, there are other pivotal factors that play significant roles in achieving convergence. Therefore, we adopt $k$-step propagation for all nodes and analyze the varying propagation states from a global perspective to obtain the Theorem 1.

**THEOREM 1.** *The upper bound on the convergence rate of $k$-step graph propagation hinges on quantifying the discrepancy between the current state and the stable state, which is defined as*

$$||\tilde{\pi} - \pi_i(k)||_2 \leq \sqrt{\frac{2m+n}{\tilde{d}_i}} \lambda_2^k, \tag{5}$$

*where the pivotal factors in striking a balance between effective convergence and over-smoothing are the High-Deg in large-scale graphs.*

PROOF. To consider the impact of each node on the others separately, let $\pi_0 = e_i$, where $e_i$ is a one-hot vector with the $i$-th component equal to 1. According to [19], we have Lemma 1.

Eq. (4) shows $(\mathbf{P}^k e_i)_j$ symbols the $j$-th component of $\mathbf{P}^k e_i$, where the $k$-step propagation started from node $i$. We denote $\mathbf{P}^k e_i$ as $\pi_i(k)$, then have the following total convergence rate variations of node $i$

$$||\tilde{\pi} - \pi_i(k)||_2^2 = \sum_{j=1}^{n}\left(\tilde{\pi}_j - \pi_i(k)_j\right)^2 \leq \frac{\sum_{j=1}^{n}\tilde{d}_j}{\tilde{d}_i}\lambda_2^{2k}$$

$$||\tilde{\pi} - \pi_i(k)||_2 \leq \sqrt{\frac{2m+n}{\tilde{d}_i}\lambda_2^{2k}} = \sqrt{\frac{2m+n}{\tilde{d}_i}}\lambda_2^k, \tag{6}$$

where $m$ and $n$ represent the number of edges and nodes. The above inequality indicates that the factors influencing the convergence rate of propagation include the degree of the current node $i$ denoted as $\tilde{d}_i$, the second largest eigenvalue $\lambda$ determined by the propagation operator, and the number of propagation step $k$.

In addition to $k$, the first influencing factor $\tilde{d}_i$ is determined by the degree of the current node $i$. Since $\tilde{d}_i \geq 1$ (with self-loop), it has minimal influence on the upper bound of the convergence rate for Low-Deg. In contrast, $\tilde{d}_i$ is directly associated with the densely connected communities (i.e. High-Deg). This explains the greater stability of the Low-Deg shown in Fig. 1 compared to the High-Deg. Then, we delve into an in-depth analysis of $\lambda_2$, narrowing our focus to the large-scale graphs. According to [19], we have Lemma 2.

The spectral gap $\Delta_{\lambda}$ denotes the difference between the magnitudes of the two largest eigenvalues of the propagation operator $\mathbf{P}$, where $\lambda_1 = 1$. Therefore, the sparse graphs (i.e. small-scale Cora) with a small value of $d_{\mathcal{G}}$ result in a relatively large value of $\lambda_2$, indicating a faster convergence rate. Contrastingly, dense graphs (i.e. large-scale ogbn-products) with a large value of $d_{\mathcal{G}}$ yield a smaller value of $\lambda_2$, presenting a unique convergence challenge. Building upon this, we have determined that the key to achieving a delicate equilibrium between efficient convergence and mitigating over-smoothing resides within the High-Deg in large-scale graphs. □

To improve convergence efficiency in large-scale scenarios, we can tackle the problem from two perspectives (excluding $k$): (i) decreasing $\tilde{d}_i$ and (ii) amplifying $\lambda_2$. Fortunately, we found that by appropriately reducing the degrees of High-Deg—thereby eliminating redundant connections—we can achieve both goals concurrently while reducing the computational costs of existing scalable GNNs. **Masking for Correction.** From a *structure-aware* perspective, we analyze the global graph propagation through Theorem 1 and find that encoding deep graph structural information of High-Deg within large-scale graphs presents difficulties, which leads to a struggled trade-off between effective convergence diameters and over-smoothing. To break these limitations, formally, we sample a subset of nodes $\tilde{\mathcal{V}} \subset \mathcal{V}$ and mask a certain percentage of their one-hop connections with a mask token [MASK], i.e., topology indicator vector $\mathbb{I}_{[M]} \in \mathbb{R}^n$ with $\theta$-based node selection threshold. Thus, the corrected topology $[\mathbf{A}_u]$ of node $u$ can be defined as:

$$[\mathbf{A}_u] = \begin{cases} \mathbb{I}_{[M]} \odot \mathbf{A}_u & u \in \tilde{\mathcal{V}} \\ \mathbf{A}_u & u \notin \tilde{\mathcal{V}} \end{cases}. \tag{7}$$

---

**Algorithm 1** Adaptive Topology-aware Propagation

**Input:** Graph $\mathcal{G}$, mask ratio $M$, threshold $\theta$, hyperparameters $C, \epsilon$;

**Output:** Node-wise propagation operator $\tilde{\Pi}$

1: Select an appropriate $\theta$ by the truncation of $\epsilon$-based inequality $||\tilde{\pi} - \pi_i(k)||_2 \leq \epsilon$ or handcraft manner;

2: Correct the high-bias propagation to obtain $[\hat{\mathbf{A}}]$ by Eq. (7);

3: Calculate the centrality-based position encoding
$\mathbf{R}_{dg} = 1 \cdot [\mathbf{D}] \cdot \text{Diag}\left(\frac{1}{n-1}, \ldots, \frac{1}{n-1}\right)$,
$\mathbf{R}_{ev} = 1/\lambda_{\max} \cdot [\mathbf{A}] \cdot (\mathbf{u}_{11}, \ldots, \mathbf{u}_{1n})$ (Selectively);

4: Calculate the connectivity-based local topology encoding
$\mathbf{R}_{dg} = \mathbb{I}(\mathcal{N}) \cdot [\mathbf{D}] \cdot \text{Diag}\left(\frac{1}{[d]_1([d]_1-1)}, \ldots, \frac{1}{[d]_n([d]_n-1)}\right)$;

5: Get node-wise propagation kernel coefficients $\tilde{\mathbf{R}}$ by Eq. (11);

6: $\tilde{\Pi} = \sum_{i=0}^{l} w_i \cdot \left(\left[\hat{\mathbf{D}}\right]^{\tilde{\mathbf{R}}-1} [\hat{\mathbf{A}}] \left[\hat{\mathbf{D}}\right]^{-\tilde{\mathbf{R}}}\right)^i$ for each propagation step;

---

Furthermore, from a *feature-oriented* perspective, unlike features in computer vision and natural language processing (i.e. high-resolution images and rich texts), graph learning often involves sparsely informative features (e.g. one-hot vectors). In large-scale graphs, disrupted homophily assumptions of High-Deg caused by intricate topology lead to the connected neighbors diverging from the current node. Consequently, these sparsely informative High-Deg struggle to maintain their uniqueness during heterophilous message aggregation. Fortunately, Eq. (7) enhances the robustness of High-Deg by regularizing the connection of misleading messages.

## 3.2 Weight-free LNC Encoding

Based on the empirical study in Sec. 1, we highlight the influence of LNC on node predictions. As stated, LNC is based on node *features*, *positions* in the graph, and *local topological structure*. A natural solution is position encoding [9, 22, 40], which helps GNNs additionally incorporate node positions. However, such segregated encoding method could inadvertently lead to misaligned learning objectives (i.e. positions and classifications), impacting the expressive capacity of GNNs. Although attention-based graph transformers [8, 36, 42] can mitigate this, they introduce extra computational costs, particularly concerning scalability when dealing with web-scale graphs. Further analysis can be found in Sec. 4.2 and Appendix A.4-A.5.

Motivated by the Fig. 1 and $\hat{\mathbf{D}}^{r-1}\hat{\mathbf{A}}\hat{\mathbf{D}}^{-r}$ from Eq. (3), different operators, guided by the propagation kernel coefficient $r$, capture LNC from different perspectives. Specifically, Low-Deg requires smaller $r$ to avoid unnecessary normalization during aggregation, acquiring more knowledge from neighbors. High-Deg benefit from relatively larger $r$, enhancing their capacity to discern neighbors. Building upon these insights, we propose weight-free LNC encoding, which employs centrality and connectivity measures to encode node *positions* and *local topological structure* in a weight-free manner. Remarkably, this strategy seamlessly integrates into *feature*-oriented scalable GNNs' graph propagation equations and coexists harmoniously with existing NP optimization strategies. Given an undirected graph, the general node-adaptive propagation kernel coefficients can be formulated as diagonal $\mathbf{R} = \sum_{k=1}^{K} \alpha_k \mathbf{P}^k \mathbf{R}_0$, where $\mathbf{P}$ is the iteration matrix and $\mathbf{R}_0$ is the initial coefficients. We use $K = 1$ and high-bias propagation optimized $\mathbf{P} = [\mathbf{D}]$ by default.

**Centrality-based Position Encoding.** In our implementation, we employ degree and eigenvector centrality for encoding node *positions* in the graph. In terms of degree-based position encoding, nodes at the center of the network (i.e. High-Deg) indicate higher influence during propagation corresponding to larger $\tilde{r}$, where $\tilde{r}$ is the optimized propagation kernel coefficient $r$.

$$\text{Degree}\left(\alpha_1 = 1, \mathbf{R}_0 = \text{Diag}\left(\frac{1}{n-1}, \ldots, \frac{1}{n-1}\right)\right) :=$$
$$\mathbf{R}_{dg} = \alpha_1 \cdot [\mathbf{D}] \cdot \mathbf{R}_0 = \text{Diag}\left(\frac{[d]_1}{n-1}, \ldots, \frac{[d]_n}{n-1}\right). \tag{8}$$

For eigenvector-based position encoding, a node's centrality depends on its neighbors, which presents a unique spectral node *positions* in the topology. This implies that High-Deg within densely connected communities possess higher influence, yielding larger $\tilde{r}$.

$$\text{Eigenvector}\left(\alpha_1 = 1/\lambda_{\max}, \mathbf{P} = [\mathbf{A}], \mathbf{R}_0 = (\mathbf{u}_{11}, \ldots, \mathbf{u}_{1n})\right) :=$$
$$\mathbf{R}_{ev} = \text{Diag}\left(\alpha_1 \cdot [\mathbf{A}] \cdot \mathbf{R}_0\right) = \text{Diag}\left(\frac{1}{\lambda_{\max}} \cdot [\mathbf{A}] \cdot (\mathbf{u}_{11}, \ldots, \mathbf{u}_{1n})\right), \tag{9}$$

where the vector $\mathbf{R}_0$ is the eigenvector corresponding to the largest eigenvalue $\lambda_{\max}$ of the optimized adjacency matrix $[\mathbf{A}]$. Based on the $\mathbf{R}_{ev}$, High-Deg pulls $\tilde{r} - 1$ closer to 0 to discern neighbors for message aggregation, while Low-Deg pushes $\tilde{r} - 1$ towards -1 to acquire more neighbor knowledge. According to $\hat{\mathbf{D}}^{\tilde{r}-1}\hat{\mathbf{A}}\hat{\mathbf{D}}^{-\tilde{r}}$ from Eq.(3), these trends satisfy the observations outlined in Sec.1.

As widely recognized, efficiently performing accurate eigendecomposition on web-scale graphs remains an open problem. However, we have opted to include $\mathbf{R}_{ev}$ as a component in our position encoding strategy. This choice stems from the fact that eigenvectors serve as spectral representations of nodes within the topology, offering a precise depiction of a node's *position*. Furthermore, we can leverage numerical linear algebra techniques to rapidly approximate solutions with error guaranteed [44, 47, 49]. Hence, under affordable computational overhead, we propose to utilize both $\mathbf{R}_{dg}$ and $\mathbf{R}_{ev}$ to further improve performance. Alternatively, if computational constraints arise, selecting solely degree-based position encoding remains a viable option. We further discuss this in Sec. 4.

**Connectivity-based Local Topological Structure Encoding.** After that, ATP represents the *local topological structure* of each node in the graph, which closely intertwines with the connectivity of their neighbors and determines the unique propagation rules. In other words, this reveals the localized connectivity patterns, where stronger connectivity corresponds to larger $\tilde{r}$, implying more consideration of the intricate neighbors, and vice versa. For instance, in social networks, nodes often form cohesive groups characterized by a notably dense interconnection of ties. This tendency is usually higher than the average probability of a random node pair [29, 56]. Therefore, we utilize local cluster connectivity with $\alpha_1 = \mathbb{I}(\mathcal{N})$ to encode this local topological structure for each node in the graph,

$$\text{Cluster}\left(\mathbf{R}_0 = \text{Diag}\left(\frac{1}{[d]_1([d]_1 - 1)}, \ldots, \frac{1}{[d]_n([d]_n - 1)}\right)\right) :=$$
$$\mathbf{R}_{cu} = \alpha_1 \cdot [\mathbf{D}] \cdot \mathbf{R}_0 = \text{Diag}\left(\frac{[d]_1 \cdot \mathbb{I}(\mathcal{N}_1)}{[d]_1([d]_1 - 1)}, \ldots, \frac{[d]_n \cdot \mathbb{I}(\mathcal{N}_n)}{[d]_n([d]_n - 1)}\right), \tag{10}$$

where $\mathcal{N}_i$ denotes the one-hop neighbors of $i$ and indicator vectors $\mathbb{I}(\mathcal{N}_i)$ is used to compute the neighborhood connectivity of $i$, i.e., $\mathbb{I}(\mathcal{N}_i) = 2|e_{jk}|$ if $v_j, v_k \in \mathcal{N}_i, e_{jk} \in \mathcal{E}$ and $\mathbb{I}(\mathcal{N}_i) = 0$ otherwise.

**Node-adaptive Propagation Kernel.** After that, we obtain the optimized propagation kernel coefficient, which is denoted as a diagonal matrix $\tilde{\mathbf{R}} \in \mathbb{R}^{n \times n}$. Building upon this, the formal representation of node-wise propagation equation $\tilde{\Pi}$ through weight-free LNC encoding combined with Eq. (3) is as follows

$$\tilde{\mathbf{R}} = C \cdot \left( \mathbf{R}_{dg} + \mathbf{R}_{ev} + \mathbf{R}_{cu} \right),$$
$$\tilde{\Pi} = \sum_{i=0}^{l} w_i \cdot \left( \left[ \hat{\mathbf{D}} \right]^{\tilde{\mathbf{R}}-1} \left[ \hat{\mathbf{A}} \right] \left[ \hat{\mathbf{D}} \right]^{-\tilde{\mathbf{R}}} \right)^i, \tag{11}$$

where $C$ is the normalization factor, $\left[ \hat{\mathbf{A}} \right]$ is the topology with self-loop after high-bias propagation correction, and $\left[ \hat{\mathbf{D}} \right]$ is the corresponding degree matrix. Remarkably, $\tilde{\Pi}$ can be seamlessly integrated into any GNN dependent on graph propagation equations (e.g. message-passing mechanisms) while maintaining orthogonality with existing NP strategies (independent of $\mathbf{L}$, $\mathbf{H}$, and $\mathbf{X}$). Furthermore, due to ATP directly optimizing the $\tilde{\Pi}$, its positive impact on decouple-based scalable GNNs is particularly pronounced.

## 4 EXPERIMENTS

In this section, we first introduce experimental setups, including datasets, baselines, and environments. More details can be found in Appendix A.1-A.3. We aim to answer the following questions to verify the effectiveness of our proposed ATP: **Q1**: How does ATP perform in improving backbones? Meanwhile, can ATP coexist harmoniously with existing NP optimization strategies? **Q2**: If ATP is effective, what contributes to its performance gain for backbones? **Q3**: If we insert ATP into the backbone, how does it affect the running efficiency? **Q4**: Compared to other NP optimization strategies, how does ATP perform when applied to sparse web-scale graphs?

### 4.1 Experimental Setup

**Datasets.** We evaluate the performance of ATP under both transductive and inductive settings. For transductive settings, we conduct experiments on citation networks (Cora, Citeseer, PubMed) [63], user-item datasets (Amazon Computer, Amazon Photo), co-author datasets (Coauthor CS, Coauthor Physics) [46], and OGB datasets (ogbn-arxiv, ogbn-products, ogbn-papers100M) [30]. For inductive settings, we perform experiments on Flickr and Reddit [67]. The statistics and description details are summarized in Appendix A.1.
**Baselines.** We conduct experiments using the following backbone GNNs: (i) Representative full-batch GNNs: GCN [38], GAT [51], GCNII [15], GATv2 [3]. (ii) Sampling-based GNNs: GraphSAGE [28], Cluster-GCN [17], GraphSAINT [67], ShaDow [66]. (iii) Decouple-based GNNs: SGC [57], APPNP [39], PPRGo [2], GBP [14], SIGN [26], $S^2$GC [76], AGP [53], GAMLP [71], GRAND+ [24]. Based on this, we compare ATP with existing NP optimization strategies, including DGC [55], NDLS [70], NDM [32], and SCARA [41]. Although GDC [27] and DGC both utilize heat diffusion function, we exclusively focus on DGC, which exhibits superior performance. To alleviate the randomness and ensure a fair comparison, we repeat each experiment 10 times for unbiased performance. Unless otherwise stated, we adopt GAMLP as the backbone and eigenvector-based LNC. The masking threshold $\theta$ is handcrafted with all nodes ranked in the Top-10% degree and 20% random sampling of other nodes.

**Hyperparameter Settings.** The hyperparameters in the backbone GNNs and NP optimization strategies are set according to the original paper if available. Otherwise, we perform a hyperparameter search via the Optuna [1]. For our proposed ATP, we explore the optimized $\theta$ for masking mechanisms in a handcrafted manner, which contains the selection ratios in all degree-ranked connected densely nodes (Top-1%-20%) and the sampling ratio range for other relatively sparse nodes is 0-0.5. The mask token $[M]$ and the normalization factor $C$ are explored within the ranges of 0 to 1.

### 4.2 Performance Comparison

**Backbone Improvement.** To answer **Q1**, we present ATP's optimization results for full-batch GNNs, sampling, and decouple-based scalable GNNs in Tables 1 and 2. Improvements highlighted in blue and red demonstrate the impressive performance of ATP as a plug-and-play NP optimization strategy. Building upon this, we observe that ATP's performance improvement is more pronounced in large-scale graphs compared to small-scale graphs. This is attributed to the fact that in large-scale graphs, ATP's propagation correction strategy masks more potential high-bias edges and LNC encoding allows for finer-grained exploration of intricate topology.

**Compared to Weighted Aggregation.** In Table 1, GraphSAGE, GAT, and GATv2 adopt well-known attention mechanisms based on edges for weighted message aggregation (marked with *). This node-pairs attention strategy is an additional alternative solution to graph propagation equation (i.e. $\Pi$ in Eq. (3)), making ATP cannot coexist with these learnable aggregation strategy. It's worth noting that although these attention-based approaches intuitively have the potential for better predictive performance, their limited receptive fields due to first-order aggregation and the modeling complexity imposed by intricate topologies often restrict their competitive performance and scalability when dealing with web-scale graphs (i.e. out-of-memory (OOM) error). Further detailed discuss about attention methods and ATP can be found in Appendix A.4-A.5.
**Compared to Existing NP Optimization Strategies.** To answer **Q1** from the perspective of generalizability, we provide performance gains brought by different NP optimization strategies for backbones in Table 3 under both transductive and inductive settings. We observe that ATP consistently produces competitive results in the context of large-scale graph learning, thereby validating the claims made in Sec. 1 that integrating high-bias propagation correction and LNC encoding can improve the comprehension of intricate topologies. Meanwhile, SIGN⋆ and $S^2$GC⋆ represent the best results of integrating ATP with SCARA and NDM optimization strategies by Eq. (3). We observe impressive results in their combination, validating that ATP coexists harmoniously with existing methods.

### 4.3 Ablation Study and In-depth Analysis

To answer **Q2**, we investigate the contributions of high-bias propagation correction (HPC), LNC encoding (LNC), and eigenvector-based position encoding (Eigen) within LNC to ATP, which is shown in Table 4. In addition, we provide an in-depth analysis for them.

**High-bias Propagation Correction.** For HPC, it is applied to reduce potential high-bias propagation through masking mechanisms. Its primary goal is to improve running efficiency, reflected in performance gains and reduced computational costs (see Sec. 3.1).

**Table 1: Model performance. The blue and red colors are the improvement of small- and large-scale datasets from ATP.**

| Type | Models | Cora | CiteSeer | PubMed | Photo | Computer | CS | Physics | arxiv | products | papers100M | Improv. |
|---|---|---|---|---|---|---|---|---|---|---|---|---|
| Full-batch GNNs | GCN | 81.8±0.5 | 70.8±0.5 | 79.6±0.4 | 91.2±0.6 | 82.4±0.4 | 90.7±0.2 | 92.4±0.8 | 71.9±0.2 | 76.6±0.2 | OOM | ↑1.86% |
| | GCN+ATP | 83.7±0.4 | 72.6±0.5 | 81.0±0.3 | 92.6±0.5 | 83.8±0.4 | 92.2±0.2 | 93.9±0.7 | 74.5±0.2 | 80.3±0.2 | OOM | ⇑4.22% |
| | GCNII | 83.2±0.5 | 72.0±0.6 | 79.8±0.4 | 91.5±0.8 | 82.6±0.5 | 91.0±0.3 | 92.8±1.2 | 72.7±0.3 | 79.4±0.4 | OOM | ↑1.71% |
| | GCNII+ATP | **84.6±0.6** | 73.2±0.5 | **81.6±0.5** | **92.8±0.7** | 83.8±0.4 | **92.8±0.2** | **94.3±1.0** | 75.4±0.2 | 83.5±0.3 | OOM | ⇑4.45% |
| | *GAT | 82.2±0.7 | 71.3±0.7 | 79.4±0.5 | 91.0±0.8 | 81.8±0.5 | 90.2±0.3 | 91.8±1.0 | 71.5±0.1 | OOM | OOM | - |
| | *GATv2 | 82.8±0.8 | 71.5±0.8 | 79.3±0.4 | 91.5±0.6 | 82.5±0.5 | 91.3±0.4 | 92.2±1.1 | 72.8±0.2 | OOM | OOM | - |
| Sampling GNNs | *GraphSAGE | 81.0±0.6 | 70.5±0.7 | 79.2±0.6 | 89.7±0.8 | 81.2±0.6 | 90.5±0.4 | 91.5±1.0 | 71.3±0.4 | 78.5±0.1 | OOM | - |
| | Cluster-GCN | 81.6±0.5 | 71.1±0.6 | 79.3±0.4 | 90.8±0.7 | 82.2±0.5 | 90.8±0.3 | 91.8±1.1 | 71.5±0.3 | 78.8±0.2 | OOM | ↑1.94% |
| | Cluster-GCN+ATP | 83.4±0.6 | **73.5±0.5** | 80.7±0.5 | 92.3±0.6 | 83.8±0.5 | 92.0±0.3 | 93.4±1.0 | 74.4±0.3 | 83.6±0.2 | 66.4±0.2 | ⇑5.07% |
| | GraphSAINT | 81.3±0.5 | 71.5±0.6 | 79.3±0.5 | 90.5±0.8 | 81.6±0.5 | 90.4±0.3 | 92.0±1.2 | 71.9±0.3 | 80.3±0.3 | OOM | ↑2.06% |
| | GraphSAINT+ATP | 83.5±0.5 | 73.3±0.7 | 81.0±0.4 | 92.0±0.7 | 83.6±0.6 | 91.8±0.2 | 93.6±1.0 | 74.9±0.2 | 83.7±0.4 | 67.2±0.2 | ⇑4.20% |
| | ShaDow | 81.4±0.7 | 71.6±0.5 | 79.6±0.5 | 90.8±0.9 | 82.0±0.6 | 91.0±0.3 | 92.2±1.0 | 72.1±0.2 | 80.6±0.1 | 67.1±0.2 | ↑2.14% |
| | ShaDow+ATP | 83.8±0.8 | 73.4±0.6 | 81.3±0.5 | 92.4±0.8 | **84.0±0.5** | 92.5±0.3 | 93.6±0.8 | **75.8±0.2** | **84.8±0.2** | **69.8±0.1** | ⇑4.38% |

**Table 2: Model performance on decoupled GNNs.**

| Models | arxiv | prodcuts | papers100M | Improv. |
|---|---|---|---|---|
| SGC | 71.84±0.26 | 75.92±0.07 | 64.38±0.15 | ⇑4.48% |
| SGC+ATP | 74.47±0.21 | 82.06±0.10 | 67.25±0.12 | |
| APPNP | 72.34±0.24 | 78.84±0.09 | 65.26±0.18 | ⇑4.51% |
| APPNP+ATP | 75.16±0.27 | 83.58±0.12 | 69.33±0.15 | |
| PPRGo | 72.01±0.18 | 78.45±0.16 | 65.87±0.20 | ⇑4.60% |
| PPRGo+ATP | 74.56±0.24 | 83.88±0.12 | 69.45±0.16 | |
| GBP | 72.13±0.25 | 78.49±0.15 | 64.10±0.18 | ⇑4.30% |
| GBP+ATP | 74.96±0.22 | 83.66±0.20 | 68.78±0.12 | |
| AGP | 72.45±0.20 | 78.34±0.13 | 65.53±0.15 | ⇑4.55% |
| AGP+ATP | 75.08±0.16 | 83.58±0.16 | 69.16±0.18 | |
| GRAND+ | 73.86±0.28 | 79.55±0.20 | 66.86±0.17 | ⇑4.24% |
| GRAND++ATP | 75.69±0.25 | 84.70±0.14 | 70.27±0.24 | |

**Table 3: Model performance with NP optimization strategies.**

| Model | products | papers100M | Flickr | Reddit |
|---|---|---|---|---|
| SIGN | 79.26±0.1 | 65.34±0.2 | 52.46±0.1 | 93.41±0.0 |
| SIGN+DGC | 82.16±0.2 | 67.84±0.2 | 53.32±0.1 | 94.92±0.1 |
| SIGN+NDLS | 81.92±0.1 | 68.10±0.1 | 53.74±0.1 | 94.58±0.0 |
| SIGN+NDM | 82.48±0.2 | 68.45±0.1 | 53.95±0.1 | 95.32±0.1 |
| SIGN+SCARA | 82.20±0.2 | 67.91±0.2 | **54.18±0.2** | 94.64±0.1 |
| SIGN+ATP | **83.65±0.1** | **68.70±0.2** | 54.06±0.1 | **95.54±0.0** |
| SIGN★ | 83.95±0.2 | 69.24±0.2 | 54.83±0.2 | 96.08±0.1 |
| S²GC | 78.84±0.1 | 65.15±0.1 | 52.10±0.1 | 92.14±0.0 |
| S²GC+DGC | 81.75±0.1 | 67.42±0.2 | 53.24±0.1 | 94.22±0.1 |
| S²GC+NDLS | 82.18±0.2 | 67.86±0.1 | 53.68±0.1 | 94.10±0.1 |
| S²GC+NDM | 82.84±0.2 | **68.20±0.2** | 54.02±0.2 | 94.86±0.1 |
| S²GC+SCARA | 82.76±0.2 | 68.04±0.2 | 54.25±0.1 | 94.57±0.1 |
| S²GC+ATP | **82.32±0.1** | 68.10±0.1 | **54.48±0.1** | **95.28±0.0** |
| S²GC★ | 83.68±0.2 | 68.87±0.2 | 55.16±0.2 | 96.18±0.1 |

**Table 4: Ablation on transductive and inductive settings.**

| Model | arxiv | products | Flickr | Reddit |
|---|---|---|---|---|
| GCNII | 72.74±0.3 | 79.43±0.4 | 53.11±0.1 | 93.65±0.1 |
| GCNII+ATP | 75.42±0.2 | 83.51±0.3 | 53.96±0.1 | 95.04±0.0 |
| w/o HPC | 75.04±0.4 | 82.97±0.4 | 53.68±0.2 | 94.76±0.1 |
| w/o Eigen | 74.83±0.3 | 82.72±0.3 | 53.55±0.1 | 94.65±0.0 |
| w/o LNC | 73.66±0.2 | 80.63±0.2 | 53.42±0.1 | 94.14±0.0 |
| ShaDow | 72.13±0.2 | 80.64±0.3 | 52.71±0.2 | 94.10±0.0 |
| ShaDow+ATP | 75.84±0.2 | 84.80±0.2 | 53.80±0.1 | 95.49±0.0 |
| w/o HPC | 75.37±0.3 | 84.04±0.2 | 53.48±0.2 | 95.02±0.1 |
| w/o Eigen | 75.04±0.2 | 83.85±0.3 | 53.28±0.1 | 94.84±0.1 |
| w/o LNC | 73.03±0.2 | 81.53±0.2 | 52.95±0.1 | 94.41±0.0 |
| GAMLP | 73.43±0.3 | 81.41±0.2 | 53.86±0.2 | 94.25±0.1 |
| GAMLP+ATP | 76.22±0.2 | 85.64±0.2 | 55.64±0.1 | 95.88±0.0 |
| w/o HPC | 75.73±0.3 | 84.96±0.3 | 54.85±0.2 | 95.50±0.1 |
| w/o Eigen | 75.69±0.2 | 84.85±0.3 | 54.47±0.1 | 95.33±0.1 |
| w/o LNC | 74.23±0.2 | 82.64±0.2 | 54.05±0.1 | 94.86±0.0 |

Building upon this, we further analyze HPC by the selection ratio of High-Deg for masking in Fig. 2. The experimental results indicate that as the masking rate increases from Top-1%, there is a consistent improvement in performance. In most cases, we suggest that select nodes with degrees in the Top-10%-15% of the degree ranking (from high to low) for masking. Excessive masking nodes may have a negative impact on predictions due to broken topology. More results and discuss about HPC can be found in Appendix A.6.

**Local Node Context Encoding.** As mentioned in Sec.1, we aim to customize propagation rules for each node in large-scale graphs with intricate topologies, while adhering to the $\Pi$ in Eq.(3). The key insight is to focus on LNC composed of node *features*, *positions* in the graph, and *local topological structure*, as it possesses unique prompts that aid the model in node-level classification downstream task. Experimental results in Table 4 confirm our claims, for instance, LNC helps improve ShaDow's performance on ogbn-products from 81.53 to 84.80. Moreover, Eigen, as a fine-grained position encoding in the spectral domain, plays a significant role in performance gains. Therefore, we suggest incorporating Eigen as part of LNC encoding, with acceptable additional computational overhead (see Sec. 3.2).

Therefore, HPC not only achieves an average improvement of 0.48% but also offers a solution for enhancing model scalability. For instance, in Table 1, HPC makes Cluster-GCN and GraphSAINT trainable on ogbn-papers100M. More details can be found in Sec. 4.4.

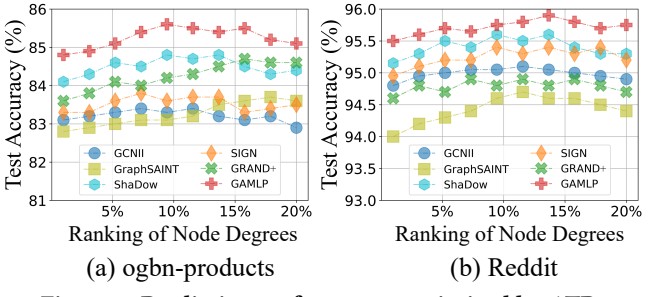

Figure 2: Predictive performance optimized by ATP.

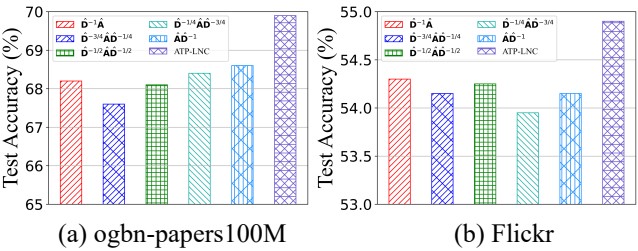

Figure 3: Predictive performance with different kernels.

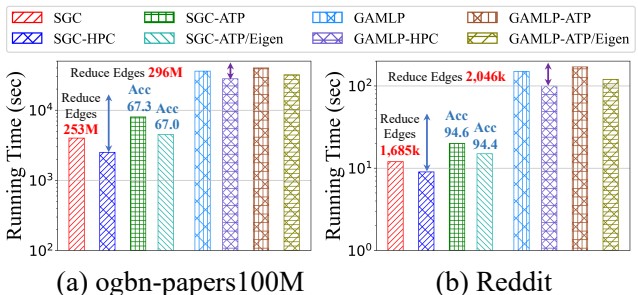

Figure 4: Running times on large-scale graphs.

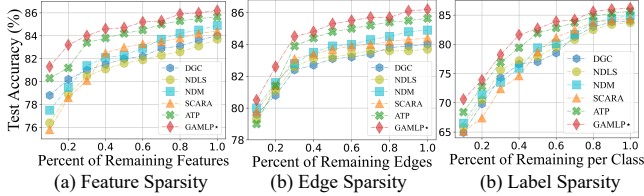

Figure 5: Sparsity performance on ogb-products.

18.7% and 17.6%. This is because GAMLP utilizes a receptive field-based attention mechanism, which reduces its reliance on intricate topologies. Therefore, in some cases, overly dense connections can cause negative impacts. (ii) While LNC introduces additional pre-processing overhead, when not using Eigen, HPC further optimizes the running efficiency for computationally complex scalable GNNs such as GAMLP. Remarkably, lightweight LNC encoding strategies (i.e. without Eigen) still exhibit robustness and competitive results (67.3%-67.0% and 94.6%-94.4%) as a plug-and-play approach.

The above observations highlight that ATP can strike a balance between model running efficiency and predictive performance through $\theta$-based masking mechanisms and selective LNC encoding strategies. This observation strongly underscores the exceptional scalability of ATP and its ability to handle web-scale graphs.

### 4.5 Performance under Sparse Graphs

To answer **Q4**, the experimental results are presented in Fig. 5. For stimulating feature sparsity, we assume that the feature representation of unlabeled nodes is partially missing. In this case, NP optimization strategies that rely on node representations like NDLS and feature-push operations like SCARA are severely compromised. Conversely, methods based on topology like heat diffusion such as DGC and NDM, along with LNC encoding, exhibit robustness. To simulate edge sparsity, we randomly remove a fixed percentage of edges. Notably, since all NP optimization strategies rely on the topology to custom propagation rules, their performance is not optimistic under the edge sparsity setting. However, we observe that ATP quickly recovers and exhibits with leading performance. For stimulating label sparsity, we change the number of labeled samples for each class and acquire the testing results with similar trend as the feature-sparsity tests. Furthermore, the performance of GAMLP★ in Fig. 5 once again demonstrates the positive coexistence effect when seamlessly integrating our proposed ATP into other optimization methods. Therefore, ATP comprehensively enhance both the performance and robustness of the original backbone.

## 5 CONCLUSION

In this paper, we first provide a valuable empirical study that reveals the uniqueness of intricate topology in web-scale graphs. Then, we propose ATP, a plug-and-play NP optimization strategy that can be seamlessly integrated into most GNNs to improve running efficiency, reflected in performance gains and lower costs.

ATP aims to address scalability and adaptability challenges encountered by existing GNNs when being implemented in complex web-scale graphs with intricate topologies. To further improve performance, finer-grained HPC can be considered, such as identifying edges (i.e. homophily or heterophily). Discovering isomorphism- and kernel-based LNC encoding are promising directions as well.

To further validate the effectiveness of LNC, we provide experimental results in Fig. 3 using different propagation kernel coefficients. We observe that, in ogbn-papers100M consisting more High-Deg, larger values of $r$ yield better results in general. Conversely, in Flickr, smaller values of $r$ are recommended. In both cases, the advantage of ATP-LNC is significant. Specifically, ATP employs node-adaptive LNC encoding to capture topological distinctions between nodes situated in different densities, thereby guiding the NP process and achieving significant predictive performance.

### 4.4 Running Efficiency in Large-scale Graphs

To answer **Q3**, we present a visualization illustrating the impact of ATP as a plug-and-play NP strategy on running efficiency for backbone GNN in Fig. 4. In our report, the running efficiency encompasses both topology-related pre-processing and model training time, the red text corresponds to the reduction in the number of edges achieved by HPC, while the blue text represents the performance influenced by Eigen. Based on this, we draw the following conclusions: (i) HPC improves running efficiency by directly reducing potential high-bias edges. For instance, in SGC, HPC reduces edges by 15.3% and 14.5% on ogbn-papers100M and Reddit. However, for GAMLP, the optimized proportion of masked edges increases to

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

# A   OUTLINE

The appendix is organized as follows:

## A.1   Dataset Description

The description of all datasets is listed below:

**Cora**, **CiteSeer**, and **PubMed** [63] are three citation network datasets, where nodes represent papers and edges represent citation relationships. The node features are word vectors, where each element indicates the presence or absence of each word in the paper.

**Coauthor CS** and **Coauthor Physics** [46] are co-authorship graphs based on the Microsoft Academic Graph [54], where nodes are authors, edges are co-author relationships, node features represent paper keywords, and labels indicate the research field.

**Amazon Photo** and **Amazon Computers** [46] are segments of the Amazon co-purchase graph, where nodes represent items and edges represent that two goods are frequently bought together. Given product reviews as bag-of-words node features.

**ogbn-arxiv** and **ogbn-papers100M** [30] are two citation graphs indexed by MAG [54]. Each paper comes with a 128-dimensional feature vector obtained by averaging the embeddings of words in its title and abstract. The embeddings of individual words are computed by running the skip-gram model.

**ogbn-products** [30] is a co-purchasing network, where nodes represent products and edges represent that two products are frequently bought together. Node features are generated by extracting bag-of-words features from the product descriptions followed by a Principal Component Analysis to reduce the dimension to 100.

**Flickr** [67] dataset originates from the SNAP, where nodes represent images, and connected images from common properties. Node features are 500-dimensional bag-of-words representations.

**Reddit** [28] dataset collected from Reddit, where 50 large communities have been sampled to build a post-to-post graph, connecting posts if the same user comments on both. For features, off-the-shelf 300-dimensional GloVe vectors are used.

## A.2   Compared Baselines

To evaluate the effectiveness of different NP optimization strategies, we utilize representative and scalable GNNs as benchmarks to establish baseline performance.

**GCN** [38] introduces a novel approach to graphs that is based on a first-order approximation of spectral convolutions on graphs. This approach learns hidden layer representations that encode both local graph structure and features of nodes.

**GAT** [51] utilizes attention mechanisms to quantify the importance of neighbors for message aggregation. This strategy enables implicitly specifying different weights to different nodes in a neighborhood, without depending on the graph structure upfront.

**GCNII** [15] incorporates initial residual and identity mapping. Theoretical and empirical evidence is presented to demonstrate how these techniques alleviate the over-smoothing problem.

**GATv2** [3] introduces a variant with dynamic graph attention mechanisms to improve GAT. This strategy provides better node representation capabilities and enhanced robustness when dealing with graph structure as well as node attribute noise.

**GraphSAGE** [28] leverages neighbor node attribute information to efficiently generate representations. This method introduces a general inductive framework that leverages node feature information to generate node embeddings for previously unseen data.

**Cluster-GCN** [17] is designed for training with stochastic gradient descent (SGD) by leveraging the graph clustering structure. At each step, it samples a block of nodes that associate with a dense subgraph identified by a graph clustering algorithm, and restricts the neighborhood search within this subgraph.

**GraphSAINT** [67] is a inductive framework that enhances training efficiency through graph sampling. Each iteration, a complete GCN is built from the properly sampled subgraph, which decouples the sampling from the forward and backward propagation.

**ShaDow** [66] decouples the depth and scope of GNNs for informative representations in node classfication. This approach propose a design principle to decouple the depth and scope of GNNs – to generate representation of a target entity, where a properly extracted subgraph consists of a small number of critical neighbors, while excluding irrelevant ones.

**SGC** [57] simplifies GCN by removing non-linearities and collapsing weight matrices between consecutive layers. Theoretical analysis show that the simplified model corresponds to a fixed low-pass filter followed by a linear classifier.

**APPNP** [39] leverages the connection between GCN and PageRank to develop an enhanced propagation method. This strategy leverages a large, adjustable neighborhood for classification and can be easily combined with any neural network.

**PPRGo** [2] proposes an efficient approximation of diffusion in GNNs for substantial speed improvements and better performance. This approach utilizes an efficient approximation of information diffusion in GNNs resulting in significant speed gains while maintaining competitive performance.

**SIGN** [26] introduces a novel, efficient, and scalable graph deep learning architecture that eliminates the need for graph sampling. This method sidesteps the need for graph sampling by using graph convolutional filters of different size that are amenable to efficient pre-computation, allowing extremely fast training and inference.

**$S^2GC$** [76] introduces a modified Markov Diffusion Kernel for GCN, which strikes a balance between low- and high-pass filters to capture the global and local contexts of each node.

**GBP** [14] introduces a scalable GNN that employs a localized bidirectional propagation process involving both feature vectors and the nodes involved in training and testing. Theoretical analysis shows that GBP is the first method that achieves sub-linear time complexity for both the pre-computation and the training phases.

**AGP** [53] proposes a unified randomized algorithm capable of computing various proximity queries and facilitating propagation. This method provides a theoretical bounded error guarantee and runs in almost optimal time complexity.

**GAMLP** [71] is designed to capture the inherent correlations between different scales of graph knowledge to break the limitations of the enormous size and high sparsity level of graphs hinder their applications under industrial scenarios.

**Table 5: The statistical information of the experimental datasets.**

| Dataset | #Nodes | #Features | #Edges | #Classes | #Train/Val/Test | #Task | Description |
|---------|--------|-----------|--------|----------|-----------------|-------|-------------|
| Cora | 2,708 | 1,433 | 5,429 | 7 | 140/500/1000 | Transductive | citation network |
| CiteSeer | 3,327 | 3,703 | 4,732 | 6 | 120/500/1000 | Transductive | citation network |
| PubMed | 19,717 | 500 | 44,338 | 3 | 60/500/1000 | Transductive | citation network |
| Amazon Photo | 7,487 | 745 | 119,043 | 8 | 160/240/7,087 | Transductive | co-purchase graph |
| Amazon Computer | 13,381 | 767 | 245,778 | 10 | 200/300/12,881 | Transductive | co-purchase graph |
| Coauthor CS | 18,333 | 6,805 | 81,894 | 15 | 300/450/17,583 | Transductive | co-authorship graph |
| Coauthor Physics | 34,493 | 8,415 | 247,962 | 5 | 100/150/34,243 | Transductive | co-authorship graph |
| ogbn-arxiv | 169,343 | 128 | 2,315,598 | 40 | 91k/30k/48k | Transductive | citation network |
| ogbn-products | 2,449,029 | 100 | 61,859,140 | 47 | 196k/49k/2204k | Transductive | co-purchase graph |
| ogbn-papers100M | 111,059,956 | 128 | 1,615,685,872 | 172 | 1200k/200k/146k | Transductive | citation network |
| Flickr | 89,250 | 500 | 899,756 | 7 | 44k/22k/22k | Inductive | image network |
| Reddit | 232,965 | 602 | 11,606,919 | 41 | 155k/23k/54k | Inductive | social network |

**GRAND+** [24] develops the generalized forward push algorithm called GFPush, which is utilized for graph augmentation in a mini-batch fashion. Both the low time and space complexities of GFPush enable GRAND+ to efficiently scale to large graphs.

## A.3 Experiment Environment

Experiments are conducted with Intel(R) Xeon(R) CPU E5-2686 v4 @ 2.30GHz, and a single NVIDIA GeForce RTX 3090 with 24GB GPU memory. The operating system of the machine is Ubuntu 20.04.5 with 768GB of memory. As for software versions, we use Python 3.8.10, Pytorch 1.13.0, and CUDA 11.7.0.

## A.4 Weighted Aggregation and LNC Encoding

In Sec. 4.2, we provide a brief discussion of potential scalability concerns associated with the attention mechanisms in web-scale graph learning scenarios. We also highlight the incompatibility of these end-to-end learnable message aggregation mechanisms with the LNC encoding introduced in ATP for optimizing propagation kernel coefficients $r$. To delve deeper into these statements and present a comprehensive evaluation about weighted message aggregation and LNC encoding within ATP, this section begins by clarifying the distinctions and connections between learnable attention mechanisms and graph propagation equations. Then, we present visual experimental results for both ATP and end-to-end attention-based approaches, including GraphSAGE [28], GAT [51], and GATv2 [3], on the ogbn-arxiv dataset for deeper analysis.

**Graph Attention Mechanisms.** To improve the predictive performance of the current node $i$, GraphSAGE proposes to explicitly consider its first-order neighbors, denoted as $j \in \mathcal{N}_i$. Specifically, during the message aggregation, GraphSAGE treats all neighbors with equal importance (indiscriminated aggregation), with a key aspect being the combination of aggregated messages using an end-to-end learnable mechanism. This can be formalized as follows

$$\mathbf{X}_u = \text{Aggregate}\left(\mathbf{W}, \mathbf{X}_u, \{\mathbf{X}_v, \forall v \in \mathcal{N}(u)\}\right). \quad (12)$$

To achieve fine-grained message aggregation for each node, GAT employs a $d$-dimension embedding-based learnable scoring function with trainable $\mathbf{a}$, $\mathbf{W}$, denoted as $e : \mathbb{R}_q^d \times \mathbb{R}_v^d \to \mathbb{R}$, to obtain

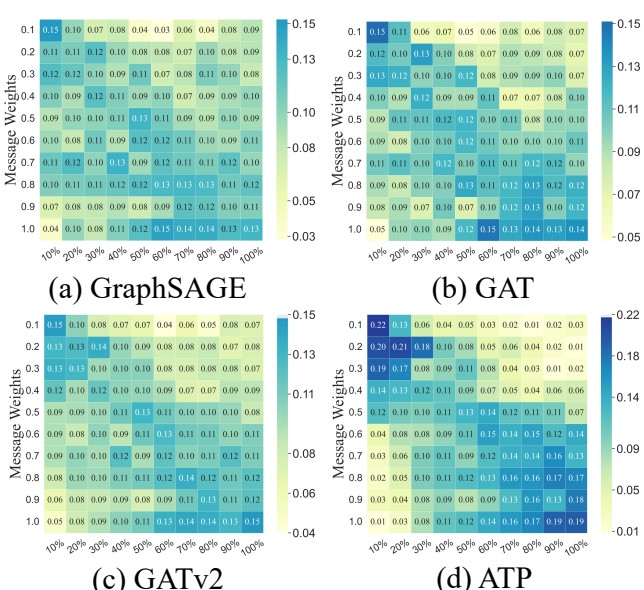

(a) GraphSAGE (b) GAT

(c) GATv2 (d) ATP

**Figure 6: Comparison of the attention- and LNC encoding-based message aggregation weights (similar to propagation kernel coefficients) on ogbn-arxiv. The x-axis represents the ranking of node degrees from low to high order.**

the attention score $\alpha$ of each "key" neighbor in generating representations for the current "query" node (i.e. attention mechanism).

$$e\left(\mathbf{X}_i, \mathbf{X}_j\right) = \text{LeakyReLU}\left(\mathbf{a}^\top \cdot \left[\mathbf{W}\mathbf{X}_i \| \mathbf{W}\mathbf{X}_j\right]\right),$$
$$\alpha_{ij} = \text{softmax}\left(e\left(\mathbf{X}_i, \mathbf{X}_j\right)\right) = \frac{\exp\left(e\left(\mathbf{X}_i, \mathbf{X}_j\right)\right)}{\sum_{j \in \mathcal{N}_i} \exp\left(e\left(\mathbf{X}_i, \mathbf{X}_{j'}\right)\right)}. \quad (13)$$

Then, GAT takes into account neighbor messages with varying scores when generating representations for the current node.

$$\mathbf{X}_u = \sigma\left(\sum_{j \in \mathcal{N}_i} \alpha_{ij} \cdot \mathbf{W}\mathbf{X}_j\right). \quad (14)$$

Due to the globally shared learnable parameters in the $e$, different "query" node embeddings $\mathbf{X}_u \in \mathbb{R}_q^d$ will yield the same score ranking list in extreme scenarios (e.g. complete bipartite graphs). In other

words, GAT may disproportionately focus on a fixed "key" neighbor (i.e. static attention), which contradicts the original intention of flexible attention composition. Building upon this observation, GATv2 modifies the order of importance scores computation to achieve a more expressive graph attention mechanism.

$$e\left(\mathbf{X}_i, \mathbf{X}_j\right) = \mathbf{a}^\top \text{LeakyReLU}\left(\mathbf{W} \cdot \left[\mathbf{X}_i \| \mathbf{X}_j\right]\right). \tag{15}$$

Reviewing graph attention, we find that their optimization can also be derived from a node-wise perspective. For instance, GAT aims to identify neighbors during aggregation, while GATv2 adopts fine-grained attention score modeling. Fundamentally, graph attention represents a specific instance within the broader context of graph propagation equations, customizing message aggregation strategies for each node in an end-to-end learnable manner (i.e. node-pair based $w$ in Eq. (3)). Intuitively, graph attention is effective, but the intricate topology in web-scale graphs brings unique challenges. According to Table 1, it is evident that current attention mechanisms struggle to maintain effective and consistent, let alone provide the scalability required for large-scale graph learning. To illustrate this issue, we provide the following visual analysis.

**Visual Analysis.** We report the visual results in Fig. 6, where the x-axis indicates the node set with degrees within the Top 10% of the ranking from low to high order and the heat map value represents the percentage of nodes in the set that have achieved the correspondent message weights. For ATP, the y-axis is the node-adaptive $r$. For others, we first train each model and then obtain the average attention score $\alpha$ for each node in first-order aggregation.

Building upon this, we draw the following conclusions: (1) The performance of ATP aligns with the key intuition derived from our empirical study in Sec.1. Specifically, nodes with smaller degrees tend to have smaller $r$, leading to a more inclusive knowledge acquisition from their neighbors, while nodes with larger degrees have larger $r$ to enable fine-grained discrimination of neighbors. (2) Progressing from GraphSAGE to GATv2, we observe that their optimization objectives align with the pre-process in ATP. However, as previously emphasized, when faced with intricate topologies in web-scale graphs, existing methods cannot fully capture potential structural patterns through learnable mechanisms. The lack of distinct color differentiations leads to their sub-optimal performance.

## A.5 Graph Transformers and GCNII-based ATP

We commence by revisiting graph transformer and graph attention from the perspective of the *attention* mechanism. Then, we elucidate the distinctions between ATP and graph transformer. With experimental analysis, we discuss if ATP is preferred to perform graph propagation on web-scale graphs comparing graph transformer.

**Graph Transformer Mechanisms.** From a self-attention perspective, the graph attention mechanism calculates only the first-order neighbors of the current node, while the graph transformer considers all nodes within the graph. From a message aggregation perspective, graph transformer correspond to fully connected dense graphs, while graph attention corresponds to a relatively sparse graph. In terms of structural encoding, graph transformers provide the model with high-dimensional global structural positional priors, whereas graph attention focuses more on the local neighbors.

Specifically, transformers consist of multiple transformer layers, with each comprising a self-attention module and a feed-forward

network (FFN). Considering the $l$-th transformer layer, the input features $\mathbf{H}^{(l-1)} \in \mathbb{R}^{N \times d}$ (where $\mathbf{H}^{(0)} = \mathbf{X}^{(0)}$) are initially transformed using three weight matrices $\mathbf{W}^Q \in \mathbb{R}^{d \times d_Q}, \mathbf{W}^K \in \mathbb{R}^{d \times d_K}, \mathbf{W}^V \in \mathbb{R}^{d \times d_V}$ to generate the corresponding query, key, and value matrices $Q, K, V \in \mathbb{R}^{N \times d}$. For simplicity, we assume that $d = d_K = d_Q = d_V$. The formulation of the transformer layer is then as follows:

$$\mathbf{Q} = \mathbf{H}^{(l-1)}\mathbf{W}^Q, \mathbf{K} = \mathbf{H}^{(l-1)}\mathbf{W}^K, \mathbf{V} = \mathbf{H}^{(l-1)}\mathbf{W}^V$$
$$\mathbf{B}^{(l)} = \frac{\mathbf{Q}\mathbf{K}^\top}{\sqrt{d}}, \quad \mathbf{H}^{(l)} = \text{FFN}\left(\text{softmax}\left(\mathbf{B}^{(l)}\right)\mathbf{V}\right). \tag{16}$$

Building upon this, graph transformers empower nodes to incorporate information from any other nodes in the graph, thereby overcoming the constraints of the limited receptive field. In other words, the fundamental concept behind graph transformers is to incorporate structural information into the transformer architecture in a learnable manner, facilitating node predictions.

**Related Works.** To compare the performance improvement brought by ATP to GCNII with models based on the graph transformer architecture, we summarize the key characteristics of representative graph transformers proposed in recent years as follows

LSPE [23] proposes a generic architecture to decouple node attributes and topology in a learnable manner for better performance. This method proposes to decouple structural and positional representations to learn these two essential properties.

Graphormer [65] utilizes node degree and neighborhood-based spatial centrality to combine additional topological structure information in the learnable message aggregation process.

Gophormer [75] utilizes well-designed sampled ego-graphs, introduces a proximity-enhanced transformer mechanism to capture structural biases for better aggregation. Meanwhile, this strategy considers the stability in training and testing.

LiteGT [7] introduces an efficient graph transformer architecture that incorporates sampling strategies and a multi-channel transformer mechanism with kernels for better performance.

SAT [10] employs various graph learning models to extract correlated structural information within the current node's neighborhood, including utilize graph Laplacian eigenvectors-based encoding mechanism to improve transformer architectures.

AGT [42] consists of a learnable centrality encoding strategy and a kernelized local structure encoding mechanism to extract structural patterns from the centrality and subgraph views to improve node representations for the node-level downstream tasks.

NAGphormer [11] treats each node as a sequence containing a series of tokens. For each node, NAGphormer aggregates the neighborhood features from different hops into different representations.

**Experimental Analysis.** we present the predictive performance of graph transformers and GCNII-based ATP, in Table 6. Notably, we opt for GCNII over GCN to ensure a fair comparison, as the simple computations in GCN appear obsolete compared with well-designed transformer mechanisms. Based on the results, we observe that graph transformers exhibit a significant advantage on small-scale datasets such as Computer and Physics. This is attributed to their ability to effectively capture the simple and direct topological structures. Conversely, graph transformers struggle to perform well as the dataset size grows due to the increasingly intricate topology, as evidenced by their performance on ogbn-arxiv and Flickr.

**Table 6: Model performance with NP optimization strategies.**

| Models | Computer | Physics | ogbn-arxiv | Flickr |
|--------|----------|---------|-----------|--------|
| GCNII | 82.64±0.5 | 92.78±1.2 | 72.68±0.3 | 53.11±0.1 |
| GCNII+ATP | 83.75±0.4 | **94.32±1.0** | **75.37±0.2** | **53.96±0.1** |
| GNN-LSPE | 83.34±0.5 | 93.90±1.4 | 72.96±0.3 | 52.24±0.1 |
| Graphormer | 82.95±0.6 | 93.54±1.3 | 72.35±0.3 | 51.86±0.2 |
| Gophormer | 83.10±0.5 | 93.67±1.1 | 72.60±0.2 | 52.28±0.1 |
| NAGphormer | 83.76±0.5 | 93.85±1.2 | 73.75±0.3 | 53.40±0.2 |
| LiteGT | 82.84±0.6 | 93.12±1.5 | 73.13±0.3 | 52.33±0.1 |
| SAT | 83.55±0.5 | 94.12±1.2 | 73.84±0.3 | 52.57±0.1 |
| AGT | **83.84±0.6** | 93.88±1.1 | 73.98±0.3 | 53.24±0.2 |

**Graph Attention/Transformer vs ATP.** Fundamentally, both graph attention and graph transformer share the core idea of achieving better message aggregation through end-to-end learnable mechanisms. However, graph attention pays more attention to local neighbors, whereas graph transformers aim to encode global topology. Although GATv2 and NAGphormer enhance the expressiveness of score function in graph attention and improve local representations in graph transformers, they both exhibit significant disadvantages as follows : (1) The representation capacity of end-to-end learnable mechanisms is contentious, especially in facing the intricate topology of web-scale graphs. In other words, the debate about whether they can successfully capture the LNC of each node remains uncertain. Our experiments on web-scale graphs have yielded unsatisfying results. (2) Complex model architectures with vast learnable parameters lead to scalability issue. Although NAGphormer can be trained on ogbn-papers100M with the use of sampling and mini-batch training strategies, it remains challenging to deploy and is prone to instability. It is highly sensitive to sampling results and training hyperparameters such as token size.

To address these issues, we introduce LNC, which offers a comprehensive node characterization based on *features*, *positions* in the graph, and *local topological structure*. As shown in our empirical study in Sec. 1, LNC reveals key insights for achieving robust node classification performance on web-scale graphs with intricate topology. Building upon this, we propose ATP to improve graph propagation equations Eq. (3), which seamlessly combines node *features*, *positions* (from a global perspective, centrality-based position encoding similar to graph transformer), and *local topological structure* (from a local perspective, connectivity-based local topological structure encoding similar to graph attention). Meanwhile, as a weight-free and plug-and-play strategy, ATP improves the running efficiency of the most of existing GNNs.

## A.6 Masks in High-bias Propagation Correction

HPC first samples a subset of nodes $\tilde{\mathcal{V}} \subset \mathcal{V}$. The design principles for the sampling mechanism are as follows: (1) selecting all nodes that rank in the Top-$\theta$% based on their degree when nodes are sorted from highest to lowest degree. This is to strike a balance between the optimal convergence radius and over-smoothing from a global graph propagation perspective. (2) Selecting partial nodes outside the above node degree rankings. Specifically, for other relatively sparse nodes, we perform random sampling with a fixed sampling

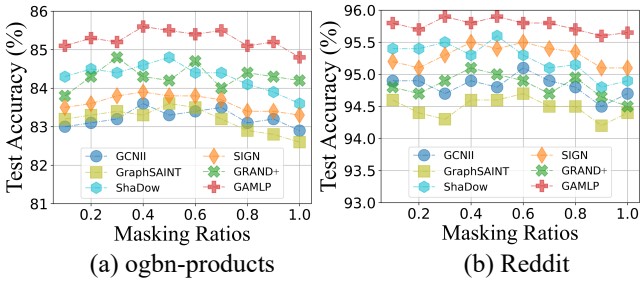

(a) ogbn-products     (b) Reddit

**Figure 7: Performance under the influence of masking ratio.**

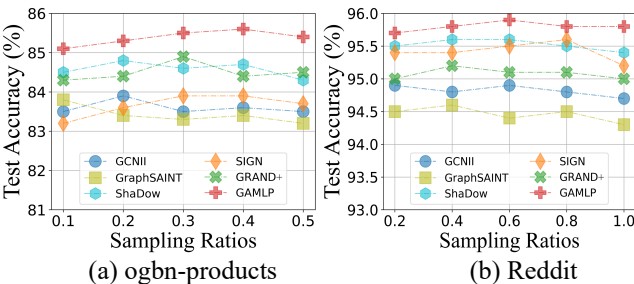

(a) ogbn-products     (b) Reddit

**Figure 8: Performance under the influence of sampling ratio.**

ratio of 0.2. Similar to dropout in training process, this strategy aims to enhance the robustness of node representations from a regularization perspective while further reducing the pre-computation and training costs associated with topology. Subsequently, HPC applies edge masking to the node set $\tilde{\mathcal{V}}$ by the mask token $[M]$. It is worth noting that we have provided experimental analysis into how different values of $\theta$ impact the performance improvement brought by High-Deg selection in Fig.2. Therefore, in this section, we supplement the discussion of the effect of the edge masking ratio $[M]$ performed by HPC on predictive performance.

According to the experimental results presented in Fig. 7, in the transductive setting, increasing the masking ratio from zero has an overall positive impact on predictive performance. However, when the masking ratio becomes excessively high, the performance deteriorates when handling edge sparsity. Furthermore, in the inductive setting, we find that lower masking ratios may have a negative effect, in stark contrast to the results in the transductive setting. The observed variation arises from the inductive setting's demand for richer neighborhood knowledge in predicting unseen nodes. Nevertheless, as we increase the masking ratio, the benefits of eliminating potential high-bias propagation outweigh the drawbacks of reduced neighborhood knowledge. In conclusion, based on the empirical analysis of the experimental results mentioned above, we recommend setting the masking ratio to 0.5, as it tends to yield optimal predictive performance in most cases.

Similar to the conclusions drawn from Fig.2 and Fig.7, the experimental results in Fig. 8 indicate that randomly sampling too many relatively sparsely connected nodes for HPC can adversely affect node prediction performance due to significant topological information gap. Conversely, selecting an appropriate sampling ratio can strike a balance between mitigating potential high-bias propagation and topological gap, thereby contributing to significant performance improvements in the original backbone.

