# OpenReview forum: "Rethinking Node-wise Propagation for Large-scale Graph Learning"
_ACM.org/TheWebConf/2024/Conference — TheWebConf24 Oral_

### Official Review · Reviewer_e9kS · 2023-11-20

**Novelty:** 5
**Technical Quality:** 5

**Review:**

This paper focuses on node classification tasks on large-scale graphs, aiming to design node-wise aggregation strategy based on local topology of nodes. Specifically, the paper presents the method Adaptive Topology-aware Propagation (ATP), which is a GNN-agnostic plugin that can be easily integrated into existing methods to improve their performance.
## Strengths：
1. This paper focuses on node-wise propagation strategy in large-scale graphs, in my opinion, this is an important and meaningful problem in web research.
2. This paper has a strong theoretical foundation, and there are no apparent issues with formula derivations or proofs.
3. Baselines are comprehensive, including traditional GNNs and decoupled GNNs, and the proposed ATP consistently improves the performance of previous methods.
4. Experiments are robust, and multiple datasets are used. Performance experiments, efficiency experiments, ablation studies etc. are conducted, demonstrating the superiority of the proposed ATP.
## Weaknesses：
1. The analysis of previous node-wise propagation strategies is insufficient, and there is no detailed explanation of why they cannot be applied to large-scale graphs.
2. I have some doubts about the results in Table 1.
  1. I am wondering why GraphSAGE experiences OOM on the papers100M dataset. In my opinion, it should be suitable for large-scale graphs by node sampling.
  2. Why does Cluster-GCN experience OOM, while Cluster-GCN+ATP does not?
3. Some of the formulas could benefit from using more easily understandable mathematical symbols in place of ':=', for example, in equation (8), (9), and etc.

**Questions:**

please see the review for more details.

**Reviewer Confidence:**

4: The reviewer is certain that the evaluation is correct and very familiar with the relevant literature

**Scope:**

4: The work is relevant to the Web and to the track, and is of broad interest to the community

---

### Official Review · Reviewer_Z2kd · 2023-11-23

**Novelty:** 6
**Technical Quality:** 5

**Review:**

GNN feature propagation suffers from its consistent scheme as well as inefficiency in computation, hindering the performance on complex and large-scale graphs. This paper investigates the two issues from a new perspective of the differences in node degrees. It presents evaluation that nodes of low and high degrees play distinct roles in affecting the feature propagation, and applying different propagation operators achieves varying performance on different graphs. Based on this observation, the paper proposes Adaptive Topology-aware Propagation (ATP) that performs High-bias Propagation Correction (HPC) and computes Local Node Context (LNC) encoding. HPC is to address the effect of node degrees in propagation. The LNC encoding based on several predefined schemes is able to represent node-wise identity, global, and local information, and can be computed with good efficiency. Extensive experiments demonstrate that the ATP framework improves performance of a set of bare-bone models while addressing the scalability issue.

This work provides an interesting perspective to understand and evaluate GNN feature propagation from graph topology, especially node degrees. It offers insightful analysis and practical implementation designs that may be helpful for large-scale GNN applications. The plug-and-play ATP framework is also applicable to a line of scalable GNNs, which increases its practical value. Comments and concerns regarding the method and implementation details are listed in below sections.

### Strengths
1. The paper rethinks the effectiveness of GNN propagation from the perspective of graph topology. It presents empirical results and theoretical analysis to show that node degrees and propagation normalization are important factors, and can be adequately addressed by HPC to improve both propagation adaptability and computational efficiency.
2. Based on the propagation analysis, the paper proposes to utilize LNC that contains multiple levels of information. The encoding is designed to be weight-free, bringing easiness and scalability for decoupled large-scale GNN computations.
3. The efficacy of the proposed ATP is evaluated on both small- and large-scale datasets, reaching up to 5% improvements. It is also demonstrated that ATP can be applied to various scalable GNN designs and improves their performance within a reasonable additional efficiency overhead.

### Weakness
1. Analysis in Section 1 and 3.1 relies on the separation of Low- and High-Deg nodes. It is however not clear on the exact separation criteria. Intuitively, expressions containing with $d_i$ such as Eq.6 usually exhibit a continuous relationship, instead of a discrete High/Low separation. See Q1.
2. Section 3.1 mentions that propagation should strike a balance between node feature convergence and over-smoothing. However, the bound in Eq.5 only corresponds to the node-wise convergence. How over-smoothing is addressed in LNC schemes? See Q2.
3. The eigendecomposition is usually known to be costly, especially for large and dense graphs. Actually in Section 4.4 and Fig.4 the additional overhead of Eigen is significant. The reliance on full-rank eigendecomposition may affects the method scalability as it aims to apply on large-scale graphs. See Q3.

**Questions:**

1. How are Low/High-Deg decided in HPC? Given that the degree distribution usually varies in different graphs, is there a general selection rule?
2. What is the effect of over-smoothing, especially for the decoupled encoding which is more sensitive to the issue? How to decide the propagation step $k$?
3. Although Section 3.2 mentions efficient calculations such as the Lanczos method, I would like to see more details regarding its performance, such as convergence and scalability evaluation. Or, is it possible to have alternative designs avoiding the direct and exact decomposition?
4. It is mentioned that the adaptive propagation scheme can be useful for fine-grained graph structures and global information. Is it possible to introduce such design to heterophilous graphs, where these information are important?
5. Efficiency evaluation in Section 4.4 and Fig.4 only compares ATP with the bare-bone models. What is the efficiency comparison between ATP and other GNNs, such as sampling-based ones? What is the computational complexity of ATP?
6. Notations such as $[A]$ and $[D]^{R-1}$ are not commonly used and should be explicitly explained.

**Reviewer Confidence:**

4: The reviewer is certain that the evaluation is correct and very familiar with the relevant literature

**Scope:**

4: The work is relevant to the Web and to the track, and is of broad interest to the community

---

### Official Review · Reviewer_cJWp · 2023-11-24

**Novelty:** 5
**Technical Quality:** 6

**Review:**

The authors propose ATP, which can be equipped with existing GNNs and Graph Transformer models without heavy modification. From the observation of the real-world datasets and theoretical analysis, the authors proposed ATP, which adaptively controls the weights of the messages depending on the characteristics of the nodes.

Pros
- The paper provides strong theoretical analysis and intuitions from observing the real-world graphs.
- GNNs and Graph Transformers equipped with ATP consistently outperform the original models and the modified models equipped with the other existing NP optimization strategies.
- The authors conducted extensive experiments to evaluate the proposed model.

Cons
- It may be hard to understand the Node-wise Propagation Optimization Strategies paragraph (from Line 294) in Section 2.2 if the readers are not familiar with NP.

**Questions:**

Q1. From the theoretical analysis, the authors suggested that reducing the degrees of high-degree nodes can mitigate the over-smoothing problem. Since the sampling strategy in GraphSAGE also reduces the degrees of high-degree nodes, it would be better to compare the proposed method with the sampling strategy from GraphSAGE (especially in Table 2).

Q2. It would be helpful if there were a mention in some parts of the main paper indicating that there are related experiments in Section A.4-6 of the supplementary material, as these experiments might be easily overlooked by readers.

**Reviewer Confidence:**

3: The reviewer is confident but not certain that the evaluation is correct

**Scope:**

3: The work is somewhat relevant to the Web and to the track, and is of narrow interest to a sub-community

---

### Official Review · Reviewer_6cHn · 2023-11-24

**Novelty:** 5
**Technical Quality:** 5

**Review:**

——————————Summary——————————

The paper addresses the challenges in scalable graph neural networks (GNNs), particularly for large-scale graph-based web applications. It identifies two main issues in existing scalable GNNs: (1) uniform propagation rules across all nodes, ignoring their topological uniqueness, and (2) insufficient node-wise propagation optimization strategies for complex web-scale graphs. To tackle these, the paper introduces Adaptive Topology-aware Propagation (ATP), a strategy that enhances node representation quality by considering the unique structural patterns of each node. This approach is designed to be integrated seamlessly into most scalable GNNs and is orthogonal to existing node-wise propagation optimization strategies.

——————————Strengths——————————

1 Providing a thorough analysis of existing works on scalable graph neural networks and highlighting their limitations is a valuable contribution for the author.

2 ATP's design as a plug-and-play solution makes it compatible with most existing scalable GNNs, enhancing its applicability.

3 The paper includes a comprehensive set of experiments conducted on various graphs, demonstrating its versatility and practicality.

——————————Weaknesses——————————

1 The focus on local node contexts might lead to overfitting in certain scenarios, especially where the global structure is also crucial.

2 The paper lacks a discussion on the theoretical computational efficiency of ATP, especially in the context of handling extremely large-scale graphs.

**Questions:**

1 How does ATP compare with other node-wise propagation optimization strategies in terms of theoretical computational efficiency and scalability?

2 Can ATP adapt to a wide range of graph topologies, including those with non-uniform node distributions or irregular structures?

**Reviewer Confidence:**

3: The reviewer is confident but not certain that the evaluation is correct

**Scope:**

4: The work is relevant to the Web and to the track, and is of broad interest to the community

---

### Official Review · Reviewer_WVbu · 2023-11-25

**Novelty:** 3
**Technical Quality:** 4

**Review:**

This work aims to develop a node-wise propagation method that is: 1) scalable and can be applied to large-scale web graphs, and 2) able to consider the unique roles played by each node in the topology. It proposes the Adaptive Topology-aware Propagation (ATP) plug-and-play module that can be incorporated into the scalable graph neural network frameworks. Essentially,  the graph convolutional filter for message passing is usually in the form of D^{r-1}AD^{-r}, and ATP decides the r by considering degree centrality, eigenvector centrality, and local topological structure (ATP sums the r solved by considering these three aspects up and then normalize). In the end, the proposed ATP is evaluated with various GNN backbones, on 12 benchmark datasets, and compared with several baseline models.



Pros:

1. The experimental results are good.
2. The authors provide extensive empirical evidence supporting the effectiveness of the ATP framework across a wide range of datasets and backbone GNNs.
3. The experimental setting is clearly given in the main paper and the supplementary.


Cons:

1. Clarity can be improved. I personally feel the paper is not very easy to follow, the authors try to deliver abundant content but the main idea is not clearly presented.
2. If I understand correctly, this work still uses the polynomial of D^{r-1}AD^{-r} to conduct message passing (formula 11 in the paper), the major novelty here is to design a non-learnable way to precompute a good r-value in this formula. In this case, the novelty seems weak.
3. The work of FSGNN (https://arxiv.org/abs/2105.07634) seems relevant and could be compared.


I appreciate the experiments in this work, the evaluation is proper and the results look good to me. However, my major concern is the novelty and clarity of this work.  Therefore, I currently would like to vote for a weak reject.

**Questions:**

Please refer to ``Review - Cons``.

**Reviewer Confidence:**

2: The reviewer is willing to defend the evaluation, but it is likely that the reviewer did not understand parts of the paper

**Scope:**

3: The work is somewhat relevant to the Web and to the track, and is of narrow interest to a sub-community

---

### Decision · Program_Chairs · 2024-01-22

**Decision:**

Accept (Oral)

**Comment:**

The paper presents a topology-aware node propagation strategy to improve efficiency and scalability in GNNs, and evaluate it across a wide range of datasets and GNNs. Reviewers appreciated the fact that this is a novel topology-aware plug-and-play approach that can be incorporated into existing scalable GNN frameworks, and the strong experimental results showcasing significant efficiency and performance gains.